# Succinct Compression: Lossless Compression for Fast and Memory-Efficient Deep Neural Network Inference

## Abstract

This paper introduces "Succinct Compression", a method to provide lossless compression of Deep Neural Network (DNN) models for fast and memory-efficient inference. The key insight of our method leverages the concept of *Succinct Data Structures*, which supports fast queries without decompressing the compressed representations. Our method consists of three new insights. First, we introduce two basic building blocks to formulate DNN models, and how they can be extended to be synergistic with compressed models (e.g. pruned or quantized models). Then, we propose a scheme to enable mixed-formulation inference for different layers, to better extract its benefits. Finally, our method exploits a specialized execution pipeline to incorporate different model formulations for fast inference. We quantitatively demonstrate that: our method can (1) enable faster and more memory-efficient inference on uncompressed models; (2) be synergistic with a variety of structure-altered/unaltered compression schemes with better speedup and compression ratio, while preserving the accuracy; and (3) can outperform all other state-of-the-art Model Coding approaches.

## 1 Introduction

Recent efforts on Pareto improvements of compressed Deep Neural Network (DNN) models, on inference time, space consumption and the accuracy, have recently bloomed due to the great success of DNNs in practice. Prior works either aggressively simplify/optimize the structure of DNN models (e.g. Pruning and Neural Architecture Search) or retrench the representation of model parameters (e.g. Quantization and Model Coding), with a major focus on the compression ratio and the accuracy. Given a variety of methodologies for efficient compression, there still lacks a general method to further optimize the inference performance and compression ratio, without affecting the accuracy of both uncompressed and compressed models.

This paper introduces "Succinct Compression", a method to provide lossless compression of Deep Neural Network (DNN) models for fast and memory-efficient inference. The emphasis of our method is to enhance the inference performance and compression ratio without affecting the accuracy at the same time, for a variety classes of uncompressed and compressed models. The unique characteristic of our method is to exploit *Succinct Data Structures*, which enables fast queries without decompressing the compressed representations.

We consolidate three new insights to better incorporate *Succinct Data Structures*. ❶ we propose two semi-structured formulations to represent DNN models in element-wise or block-wise manners, and provide simple extensions to allow them for the combinations of other compression techniques. ❷ we enable mixed formulations of different layers in the model, to better extract the potential of *Succinct Data Structures*. ❸ we design a specialized execution pipeline to perform the inference on different formulations, by carefully engineering the inner operators of *Succinct Data Structures*.

Our evaluation shows that our method can be very effective for the inference efficiency, and generally applicable for uncompressed and compressed models (including for ResNet-50, ResNet-101, VGG-16, MobileNet-V2 and DeiT-B). For uncompressed models, our method can achieves most $1.07\times$ speedup and $1.17\times$ compression ratio at the same time, without affecting the accuracy. We then show that our method can bring significantly more benefits by combining other compression

schemes, where all models are pre-processed via other compression methods. For instance, by combining structure-altered compression (such as pruning), our method enables the at most $8.8\times$ acceleration of inference on ResNet-101, with $39.90\times$ compression ratio meanwhile. Similarly, the speedup can be further enhanced to reach $9.3\times$ by incorporating structure-unaltered method (such as quantization). We also compare our method with a variety of the state-of-the-art Model Coding schemes, and show that our method outperforms all of them.

## 2 RELATED WORKS

A large body of relevant works on compressing DNN models consists of two categories, based on the orientation of their methodology: structure-altered and -unaltered methodologies. We outline key directions in each category, briefly describe their features and justify the novelty of our method.

### 2.1 STRUCTURE-ALTERED METHODS FOR COMPRESSION

**Structure-altered Methods** refer to those compression methods by simplifying/optimizing the DNN model architectures, and representative methods in this direction include Pruning, Low-Rank Factorization, Neural Architecture Search (NAS) and Knowledge Distillation (KD). We describe each of them in brief as follow.

❶ **Pruning** removes the redundant connections within DNN models without incurring a considerable degradation of the accuracy. There are two categories of Pruning. One is Unstructured Pruning (Dong et al. (2017); Lee et al. (2019); XIAO et al. (2019); Park* et al. (2020)), which aggressively removes neurons with small relevance whenever it's possible. Though such an approach can deliver decent compression ratio with only a marginal degradation of the accuracy, the inference overheads suffers from the inefficient usage of the memory, due to the frequent operations on sparse matrices (Gale et al. (2019); Blalock et al. (2020)). The other is Structured Pruning (Huang & Wang (2018); Lin et al. (2018); Yu et al. (2018); He et al. (2019); Zhao et al. (2019); Yu et al. (2021)), which only removes irrelevant units of DNN models at a granularity of the elementary structures (e.g. weights, filters and layers). Though these methods can benefit the performance/compression ratio due to the reduction of the total computational costs, the accuracy is usually not as expected.

❷ **Low-Rank Factorization** (Mamalet & Garcia (2012); Sainath et al. (2013); Zhao et al. (2017); Li et al. (2018)) uncovers the latent compact structure of the network through low-rank matrix factorization of weight layers. Though these approaches may only incur a marginal degradation in terms of the accuracy, they requires extra computational costs and the benefits in memory efficiency may not be consistent in different models.

❸ **NAS** (Mellor et al. (2021); Zhao et al. (2021)) automatically output neural network architectures using specific search strategies applied to a large search space. Therefore, a huge amount of extra computational costs are required and such methods need to be performed before the deployments of the selected models.

❹ **KD** (Feng et al. (2021); Wang (2021); Zhu et al. (2021)) is to train a large model and then use it as a teacher to train a more compact model. Similarly, KD also demands a huge amount of extra computational costs for training different models, therefore they are usually performed off-line.

In this work, we consider Pruning as the representative method in this direction, to justify the compatibility of our method with Structure-altered methods (as described in Section 7.2).

### 2.2 STRUCTURE-UNALTERED METHODS FOR COMPRESSION

**Structure-unaltered Methodologies** refer to those compression methods by compressing DNN models without altering the model architecture, and there are two representative methods in this direction, which are Quantization and Model Coding. We describe each of them in brief as follow.

❶ **Quantization** reduces the bitwidth of parameters within DNN models, and such an approach can be achieved via quantization-aware training (Bengio et al. (2013); Alizadeh et al. (2020)) or post-training quantization (Banner et al. (2019); Cai et al. (2020)). Note that it's also feasible to perform extreme quantization (e.g. binarization) for this purpose (Cai et al. (2017); Bulat et al. (2021)) but usually suffers from a significant degradation of the accuracy.

❷ **Model Coding** represents DNN models via an extra bit sequence. Several well-studied coding strategies like Huffman Coding (van Leeuwen (1976)), Tunstall Coding (Tunstall (1967)) and Arithmetic Coding (Witten et al. (1987)) have been attempted for compressing DNN models (Han et al. (2016a); Reagen et al. (2017); Zhe et al. (2021)). More recently, there are a growing interests in customized coding strategies in the context of DNN models (Louizos et al. (2017); Havasi et al. (2018); Oktay et al. (2020b)). However, these approaches suffers from costly pre-processing to convert model parameters as encoded values, and the overheads of decoding are significant during the inference runtime.

Our work considers Quantization as the representative method in this direction, to justify the synergy of our method with Structure-unaltered methods (as described in Section 7.3). Also, we compare our method against a variety of Model Coding methods to show that our method can outperform all other state-of-the-art methods (as described in Section 7.4).

## 2.3 NOVELTY OF OUR METHOD

The novelty of our method is three-folded. ❶ We are the first to introduce *Succinct Data Structures* in the context of DNN models, which allow fast queries on compressed representations. ❷ we design specific formulations to make DNN models more compatible with *Succinct Data Structures*, which reduces the overheads for both pre-processing and decoding during the inference runtime. ❸ our method is generally applicable to available DNN models, and achieve Pareto improvements in both inference time and memory efficiency, without affecting the accuracy.

## 3 FORMULATING DNN MODELS

The first part of our method is to formulate DNN models appropriately, so that *Succinct Data Structures* can take advantage of. *Succinct Data Structures* exploits the delimiters within a long string, to perform fast queries directly on the compressed representations. To this end, we propose the model formulation called Runtime-Accessible Sequence (RAS), which refers to semi-structured format using a minimal amount of delimiters to construct hierarchical information (e.g. layers). Our proposed RAS consists of two basic building blocks, including (1) Element-wise RAS, which uses delimiters to separate different elementary operands within DNN models; and (2) Block-wise RAS, which applies delimiters to separate different sets of data operands within DNN models, based on the computation kernels. Based on the above Element-wise and Block-wise RAS, we provide simple extensions of RAS, to make them synergistic with other compression methods.

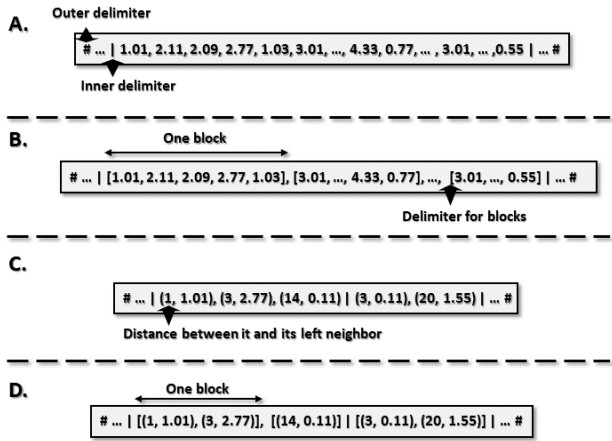

Figure 1: A comparison of different kinds of Runtime-Accessible Sequence (RAS).

### 3.1 ELEMENT-WISE RUNTIME-ACCESSIBLE SEQUENCE

One formulation in our method is Element-wise Runtime Accessible Sequence (denoted as Element-wise RAS). Element-wise RAS utilizes delimiters to separate elementary data operands. In the context of DNN models, the pre-defined delimiters (e.g. vertical bar and number sign) are used at the boundaries of different elementary data operands from DNN models, and these delimiters are used to query for elementary data operands accordingly.

Figure 1-(A) shows an example of Element-wise RAS; there are two vertical bars encompassing several elementary operands. This methodology forms the Element-wise RAS, and the number sign is used to represent the border of this union. To properly formulate the whole network into Element-wise RAS, we concatenate such unions by using a separate delimiter (e.g. '#').

### 3.2 BLOCK-WISE RUNTIME-ACCESSIBLE SEQUENCE

The limitation of Element-wise RAS is that frequent queries are required for every single data operand, before the computation for model inference. Therefore, to improve the efficiency of operand query, we suggest the other formulation of DNN models: Block-wise Runtime-Accessible Sequence (denoted as Block-wise RAS). Different from Element-wise RAS, Block-wise RAS forms basic building blocks for query and access based on the computation kernels, namely denoted as a block. Such a block stores a consecutive number of elementary data operands, which are used for a computation kernel. Between different blocks, Block-wise RAS exploits delimiters for separation, so that they can be efficiently queried.

Figure 1-(B) shows an example of Block-wise RAS: the Block-wise RAS aggregates five operands with two square brackets, as one individual block. This transformation of elementary operands, by synthesizing multiple operands and using a distinct delimiters, can provide faster queries by extracting them at one time over Element-wise RAS.

### 3.3 EXTENDING BOTH RAS FOR COMPRESSED MODELS

Since compressed models, supported by other compression schemes, usually maintain a high extent of the sparsity[1], current designs of Element/Block-wise RAS may not be capable to extract the maximum potentials of *Succinct Data Structures* on compressed models. To resolve this issue, we provide simple extensions to RAS so that they can be synergistic with other compression schemes. The key insight is to form elementary data operand in a similar manner as inverted indexes, by forming a tuple consisting of the exact values and the relative positions.

Figure 1-(C)/(D) shows examples of these optimized RAS formulated in Element- or Block-wise manner. The difference hereby is that, we refine the elementary operands as tuples. In such a tuple, the first element refers to the relative distance between this and its left neighbor (a number or a delimiter); and the second element stores the value of the corresponding data operand. This approach is synergistic with sparse models because for unstructured pruning, the relative distance, contained in the reshaped tuple, can effectively exploit the sparse model structures.

## 4 CONVERTING RAS TO *Succinct Data Structures*

The second part of our method is about how to covert RAS into *Succinct Data Structure* for efficient inference. We first provide a selection scheme to embrace the heterogeneity of different layers within DNN models, to maximize the benefits of our method. Then, we describe the rationale of our selected *Succinct Data Structure* - Wavelet Tree.

### 4.1 LAYER-WISE SELECTION OF THE RAS TYPE

In practice, soely using Element-wise or Block-wise RAS for one model cannot fully unleash the potentials, since the sparsity and access frequency vary across the whole model. Therefore, it's

---

[1]Note that Quantization and Model Coding can be viewed to exploit *bit-level* sparsity, so that the redundancy of data representation within DNN models can be reduced.

demanded to perform the selection of RAS at a certain granularity of model structure. We consider layer-level in this work, by connecting the size of elementary operands with the following criterion.

- Access Frequency of elementary data operands is used to determine which RAS to be used for the overall efficiency. More specifically, we define whether a particular layer is computationally intensive or not based on the ratio of FLOPs to #Parameters (namely FLOPs-Parameter-ratio, *FPr*). As long as above *FPr*, our method uses Block-wise RAS. Otherwise, Element-wise RAS is used.

**How RAS Types are Distributed?** We confirm that, from our experimental studies, all convolution layers and the weight matrices, inside the multi-head attention layers of Transformers, exploit Block-wise RAS, due to the intensively usage of stored data; And Element-wise RAS is applied to less-frequently-accessed layers (FC layers, in our experimental studies).

### 4.2 *Succinct Data Structures* and Wavelet Tree

*Succinct Data Structures* were first pioneered by Jacobson (1988), which refers to a set of data structures using the near-information-theoretic bound space to store the compressed representation, and still provide fast query and access operations directly on these compressed representations. In general, *Succinct Data Structures* have the following representative inner operators.

Given a string $S$ whose length and alphabet are $L$ and $\sigma$, there are three operations directly on the compression (shown below).

- $Rank_q(x)$ returns the number of symbol $q$ appearing in $S_{0:x}$ where $q \in \sigma$ and $x < L$.

- $Select_q(x)$ returns the position of $x$-th occurrence of symbol $q$ in $S$.

- $Access(x)$ returns the symbol at the position $x$ of $S$.

Though there are a number of *Succinct Data Structures* available for real-world applications, we choose Wavelet Tree (Grossi et al. (2003)) as the core of our method. We choose Wavelet Tree (WT) because there are already a number of evident successes in applying WT for large-scale, data-intensive applications, such as Data Store (Agarwal et al. (2015); Khandelwal et al. (2016)), Graph Processing (Khandelwal et al. (2017)), and it presents outstanding merits of runtime fast queries without losing memory efficiency: it allows $Rank$, $Select$ and $Access$ to only take $O(log\sigma)$ time while maintaining space consumption within $n\,log(\sigma) + O(n)$ bits (where input is of length $n$ with $\sigma$ distinct symbols). Therefore, our method deploys WT as the compression technique during the inference runtime.

## 5 Model Inference in *Succinct Data Structures*

The third part of our method is to perform model inference via *Succinct Data Structures*. There are three steps: ❶ Identify the RAS flag to guide the subsequent steps; ❷ Execute different retrieval strategies for either Element- or Block-wise RAS; and ❸ Perform the inference.

❶ **Identify the RAS Flag.** Our method first identifies the RAS flag to obtain which type of RAS being used for the current layer. The flag is tagged at the beginning of all RAS for different layers. Therefore, identifying and parsing the flag is necessary for the subsequent steps.

❷ **Execute Different Retrieval Strategies based on the Flag.** Our method then retrieves all the operands within this layer. For a given type of RAS, the retrieval strategy is structurally organized similarly: for both Element-wise and Block-wise RAS: we use two $Select$ operators to locate the corresponding values for current layer within the compressed representation, while the access of these data differs. The differences between two strategies for different types of RAS are: since Block-wise RAS retrieves a series of operands at a time, additional operations for extracting them are necessary.

❸ **Inference over the extracted operands.** Our method finally performs the inference for the current layer. Note that, for inference on compressed models, the inference may require additional decoding, if the combined methods introduce more levels of indexes.

## 6 EXPERIMENTAL METHODOLOGY

**Platform and Baseline:** All experiments are done via the latest version of PyTorch (Paszke et al. (2019b)), on a platform equipped with Intel Core i9-12900KF and NVIDIA RTX 3090. We consider five mainstream models, including ResNet-50, ResNet-101 (He et al. (2015)), VGG-16 (Simonyan & Zisserman (2015)), MobileNet-V2 (Sandler et al. (2019)) and DeiT-B (Touvron et al. (2021)).

**Methodology:** For all uncompressed and compressed models, we measure (1) the end-to-end latency to examine the performance benefits; (2) the overall memory usage to examine the memory efficiency; and (3) the accuracy. To implement the execution pipeline of our method, the inference pipeline requires less-than-byte packing and unpacking. This can be done directly using Bit Manipulation Instructions extensions (e.g. NVIDIA (a)), due to the fact that the processing granularity of modern SIMD architectures (including CPU SIMD and GPU) is at byte-level. We use PyTorch Bindings with Fortran, to exploit these hardware intrinsic functions for the implementations (i.e. Alexeev).

**Model Setups:** All these models are trained on the ImageNet dataset (Deng et al. (2009)), which are used as uncompressed models (used in Section 7.1). To examine the benefits of our method over compressed models, we apply representative Pruning (described in Section 7.2) and Quantization methods (described in Section 7.3) on the trained models, and use them as the compressed models. Furthermore, we compare our method with other Model Coding methods to show the benefits.

## 7 EXPERIMENTAL RESULTS

### 7.1 IMPACTS ON UNCOMPRESSED MODELS

We first examine the impacts of our method on uncompressed models. Figure 2 reports the model size and speedup of our method over the uncompressed models. We draw two observations. First, our method provides considerable improvement for uncompressed models on both speedup and compression. For instance, MobileNet-V2 achieves $1.07\times$ speedup and $1.17\times$ compression ratio at the same time, and ResNet-101 achieves $1.04\times$ speedup and $1.11\times$ compression ratio at the same time. Second, we note that the improvements on VGG-16 are relatively worse than others. This is because the FC layers within VGG-16 incur significant overheads for the acceleration and compression.

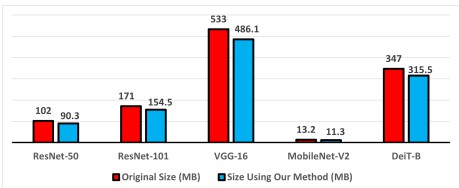 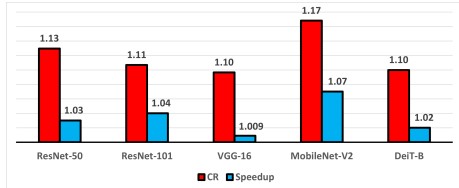

Figure 2: Results on uncompressed models for Model Size (left); CR and Speedup (right)

### 7.2 IMPACTS ON PRUNED MODELS

We then examine the impacts of our method on Pruned models. Table 1 reports the results on selected models after using three representative state-of-the-art pruning methods (i.e. layer-level pruning: DBP (Wang et al. (2019)); filter-level pruning: HRank (Lin et al. (2020)); and unstructured pruning: SNIP (Lee et al. (2020))). We first quantitatively demonstrate the outstanding impacts from our method on all models, then we elaborate the key observation.

Our method yields significantly better improvements when all models are pruned in advance, in terms of both speedup and compression ratio. The results show that our method achieves the speedup of $4.7\times/8.8\times/3.9\times/3.8\times$ and the compression ratio as $39.48\times/39.90\times/3.78\times/3.81\times$, without affecting the accuracy, for ResNet-50/ResNet-101/VGG-16/DeiT-B. Our results suggest that our method is synergistic with the state-of-the-art pruning methods.

We make the key observation that: our method yields better benefits from filter-level/unstructured pruning, compared with layer-level pruning. As shown in Table 1, the speedup from our method can be up to $1.91\times/8.07\times$ for filter-level/unstructured pruned models, but the improvement for layer-level pruned models (DBP) is limited to just $1.28\times$. This is because the filter-level/unstructured pruning decreases the "width"/parameters of each layer, and there result in less *Access* operations for extracting data operands, since these layers are formulated in Block-wise RAS based on our methodology.

Table 1: Succinct Compression on Pruned Models: Target sparsity refers to the expected sparsity within the networks after conducting pruning, and we can tolerate the error within $\pm4\%$. The entries "xx/yy" under the Size, CR (Compression Ratio) and Speedup columns show the comparison between only using pruning (left) and applying our method on the top of pruned models (right). UP-ViT (Yu & Wu (2021)) is the pruning method specialized for transformers.

| NN | M | Target Sparsity (%) | Accuracy (Top-1) (%) | Size (MB) | CR (×) | CR Improved (×) | Speedup (×) | Speedup Improved (×) |
|---|---|---|---|---|---|---|---|---|
| ResNet-50 | - | 0 | 75.5 | 102 | 1 | 1 | 1 | 1 |
| | DBP | 40 | 75.1 | 60.8/52.9 | 1.68/1.93 | 1.15 | 1.4/1.6 | 1.14 |
| | | 50 | 74.8 | 50.2/43.6 | 2.03/2.34 | 1.15 | 1.7/1.8 | 1.06 |
| | | 60 | 73.5 | 41.0/35.3 | 2.49/2.89 | 1.16 | 2.3/2.6 | 1.13 |
| | | 70 | 72.3 | 33.2/28.6 | 3.07/3.56 | 1.16 | 3.0/3.4 | 1.13 |
| | HRank | 40 | 74.4 | 61.0/53.1 | 1.67/1.92 | 1.15 | 1.3/1.9 | 1.46 |
| | | 50 | 72.5 | 49.9/43.2 | 2.04/2.36 | 1.15 | 1.4/2.6 | 1.86 |
| | | 60 | 69.9 | 41.2/35.5 | 2.48/2.87 | 1.16 | 2.1/2.7 | 1.23 |
| | | 70 | 68.1 | 31.0/26.7 | 3.29/**3.83** | 1.16 | 2.7/**4.5** | 1.67 |
| | SNIP | 90 | 74.8 | 10.3/8.7 | 9.90/11.69 | 1.18 | 1.12/2.8 | 2.50 |
| | | 95 | 74.7 | 5.2/4.4 | 19.6/23.34 | 1.19 | 1.14/3.5 | 3.07 |
| | | 97 | 74.7 | 3.1/2.6 | 32.9/**39.48** | **1.20** | 1.15/**4.7** | **4.09** |
| ResNet-101 | - | 0 | 76.4 | 171 | 1 | 1 | 1 | 1 |
| | DBP | 40 | 76.2 | 103/91.2 | 1.66/1.88 | 1.13 | 1.2/1.5 | 1.25 |
| | | 50 | 76.2 | 87/76.3 | 1.97/2.24 | 1.14 | 1.6/2.0 | 1.25 |
| | | 60 | 75.3 | 70/61.2 | 2.44/2.79 | 1.14 | 1.9/2.4 | 1.26 |
| | | 70 | 74.9 | 50.7/44.1 | 3.37/3.88 | 1.15 | 2.5/3.2 | **1.28** |
| | HRank | 40 | 76.3 | 102/90.3 | 1.68/1.89 | 1.13 | 1.1/2.1 | **1.91** |
| | | 50 | 76.1 | 85/74.7 | 2.01/2.29 | 1.14 | 1.4/2.6 | 1.86 |
| | | 60 | 75.0 | 71/62.3 | 2.41/2.75 | 1.14 | 1.8/2.9 | 1.61 |
| | | 70 | 74.3 | 50/43.4 | 3.42/**3.94** | 1.15 | 2.3/**4.4** | **1.91** |
| | SNIP | 90 | 76.0 | 17.1/14.6 | 10.0/11.7 | 1.17 | 1.08/5.3 | 4.91 |
| | | 95 | 75.9 | 8.6/7.3 | 19.9/23.56 | **1.19** | 1.09/6.4 | 5.87 |
| | | 97 | 75.9 | 5.1/4.3 | 33.5/**39.90** | **1.19** | 1.09/**8.8** | **8.07** |
| VGG-16 | - | 0 | 71.3 | 533 | 1 | 1 | 1 | 1 |
| | DBP | 40 | 71.3 | 320/290.4 | 1.67/1.84 | 1.10 | 1.7/1.9 | 1.12 |
| | | 50 | 71.3 | 262/236.0 | 2.03/2.26 | 1.11 | 2.3/2.4 | 1.04 |
| | | 60 | 71.1 | 210/189.2 | 2.54/2.82 | 1.11 | 2.7/2.9 | 1.07 |
| | | 70 | 70.2 | 158/141.1 | 3.37/**3.78** | 1.12 | 3.8/**3.9** | 1.03 |
| | HRank | 40 | 71.3 | 323/292.8 | 1.65/1.82 | 1.10 | 1.5/1.7 | 1.13 |
| | | 50 | 70.8 | 254/228.6 | 2.10/2.33 | 1.11 | 1.7/1.8 | 1.06 |
| | | 60 | 70.5 | 214/192.1 | 2.49/2.77 | 1.11 | 1.8/2.1 | 1.17 |
| | | 70 | 69.8 | 160/142.9 | 3.33/3.73 | 1.12 | 2.0/2.4 | 1.20 |
| | SNIP | 90 | 71.0 | 53/46.1 | 10.06/11.57 | 1.15 | 1.08/1.3 | 1.20 |
| | | 95 | 69.9 | 26.7/22.9 | 19.96/23.26 | **1.17** | 1.08/1.4 | 1.30 |
| | | 97 | 69.9 | 16/13.6 | 33.31/**39.08** | **1.17** | 1.1/1.7 | 1.55 |
| DeiT-B | - | 0 | 81.8 | 347 | 1 | 1 | 1 | 1 |
| | UP-ViT | 40 | 81.7 | 210/189.2 | 1.65/1.83 | 1.11 | 1.5/1.9 | 1.27 |
| | | 50 | 81.7 | 170/152.6 | 2.04/2.27 | 1.11 | 2.3/2.9 | 1.26 |
| | | 60 | 81.6 | 141/125.8 | 2.46/2.76 | 1.12 | 2.6/3.3 | 1.27 |
| | | 70 | 81.5 | 103/91.2 | 3.37/**3.81** | **1.13** | 2.9/**3.8** | **1.31** |

### 7.3 IMPACTS ON QUANTIZED MODELS

Next we examine the impacts of our method on quantized models. we use first quantize all models at a set of different levels with different bit precision (2/4/8 bits). To quantize our selected models, we use the state-of-the-art quantization methods, including LSQ (Esser et al. (2020)) for CNNs and PQT (Liu et al. (2021)) for transformer. After quantization, we apply our method upon these quantized models. We first report our quantitative results in terms of the speedup and compression ratio. Then, we elaborate the key observation.

Table 2: Succinct Compression on Quantized Models: The entries "xx/yy" under the Size, CR (Compression Ratio) and Speedup columns show the comparison between only using quantization (left) and applying our method on the top of quantized models (right). For the quantization of DeiT-B, we modify the framework PQT to output fixed-bits-length results and handcraft a DeiT-B model comprised of 2-bits parameters (marked with '*'), since PQT is incapable for the 2-bit precision.

| NN | M | Precision (bits) | Accuracy (Top-1) (%) | Size (MB) | CR (×) | CR Improved (×) | Speedup (×) | Speedup Improved (×) |
|---|---|---|---|---|---|---|---|---|
| ResNet-50 | - | 32 | 75.5 | 102 | 1 | 1 | 1 | 1 |
| | LSQ | 8 | 75.4 | 25.5/21.6 | 4.00/4.72 | 1.18 | 3.5/3.8 | 1.09 |
| | | 4 | 75.3 | 12.8/10.7 | 7.97/9.56 | 1.20 | 3.4/6.6 | 1.94 |
| | | 2 | 72.5 | 6.4/5.1 | 15.94/**19.92** | **1.25** | 2.9/**7.0** | **2.41** |
| ResNet-101 | - | 32 | 76.4 | 171 | 1 | 1 | 1 | 1 |
| | LSQ | 8 | 76.3 | 42.8/36.6 | 4.00/4.67 | 1.17 | 3.2/3.9 | 1.22 |
| | | 4 | 76.3 | 21.4/18.0 | 7.99/9.51 | 1.19 | 3.1/7.2 | 2.32 |
| | | 2 | 74.2 | 10.7/8.8 | 15.98/**19.50** | **1.22** | 2.5/**9.3** | **3.72** |
| VGG-16 | - | 32 | 71.3 | 533 | 1 | 1 | 1 | 1 |
| | LSQ | 8 | 71.3 | 134/117.5 | 3.98/4.53 | 1.14 | 3.4/3.2 | 0.94 |
| | | 4 | 71.5 | 66.6/56.9 | 8.00/9.36 | 1.17 | 3.1/3.4 | **1.10** |
| | | 2 | 69.5 | 33.3/28.0 | 16.01/**19.05** | **1.19** | 2.6/**3.8** | 1.46 |
| DeiT-B | - | 32 | 81.8 | 347 | 1 | 1 | 1 | 1 |
| | PQT | 8 | 81.3 | 86.8/75.5 | 4.00/4.60 | 1.15 | 2.6/3.3 | 1.27 |
| | | 4 | 75.9 | 43.6/36.9 | 7.96/9.39 | 1.18 | 2.4/5.2 | 2.17 |
| | * | 2 | 67.2 | 21.7/18.1 | 15.99/**19.19** | **1.20** | 1.8/**6.7** | **3.72** |

After quantization, our method can achieve the maximum level of speedup across all experiments, and still maintain a considerable amount of the reduction in terms of the memory footprint. On ResNet-101 quantized at 2-bit precision, the speedup can be achieved by 9.3×. For other quantized models at 2-bit granularity, ResNet-50/VGG-16/DeiT-B, the speedup can reach to 7.0×/3.8×/6.7×. As for the compress ratio, combining quantization and our method can further compress the models to a certain extent. For all four models selected for our experiments, our method can realize more than 19× compression ratio.

We make the key observations that: similar to our methods on Pruning, our method on quantized models also bring more benefits on CONV-dominated models, rather than FC-dominated models. For VGG-16 with large FC layers, the improvement of speedup is restricted to 1.10× at most. This is similar with the results when our method is applied on pruned models. We also report results by applying our method on DeiT-B. Though this model is distinctively different from CNNs, our method can still provide a considerable amount of space reduction and inference acceleration, by up to 19.19× compression and 6.7× speedup.

### 7.4 A COMPARISON AGAINST OTHER MODEL CODING METHODS

Finally, we compare our method with the state-of-the-art Model Coding methods, including Rate-distortion Optimized Coding (ROC) (Zhe et al. (2021)), Entropy Penalized Reparameterization (EPR) (Oktay et al. (2020a)), DeepCABAC (Wiedemann et al. (2019)), Minimal Random Code

Table 3: A comparison between Succinct Compression and other Model Coding Methods: for a fair comparison, we only consider the best results from baseline methods. ('-' means the speedup is not capable to be validated, since they are not reported in the original paper.)

| Neural Network Models | Method | Size (MB) | CR (×) | Speedup (×) | Error (Top-1) (%) |
|---|---|---|---|---|---|
| | Uncompressed | 102 | 1 | 1 | 24.5 |
| | OPQ | 2.68 | 38.0 | 1.3 | **24.5** |
| | Deep Compression | 6.41 | 15.91 | 0.8 | 25.7 |
| ResNet-50 | DeepCABAC | 6.06 | 16.8 | - | 25.9 |
| | EPR(DFT) | 5.49 | 18.6 | - | 26.0 |
| | ROC (Tunstall) | 5.10 | 20.0 | 4.3 | 24.9 |
| | **Our Method** | 2.10 | **48.56** | **9.2** | **24.5** |
| | Uncompressed | 13.2 | 1 | 1 | 29.0 |
| | OPQ | 0.57 | 23.2 | 1.2 | **29.3** |
| MobileNet-V2 | Deep Compression | 0.97 | 13.62 | 1.1 | 30.5 |
| | ROC (Tunstall) | 1.2 | 11.0 | 1.7 | 29.8 |
| | **Our Method** | 0.44 | **30.11** | **5.6** | **29.3** |
| | Uncompressed | 60 | 1 | 1 | 6.6 |
| | OPQ | 0.13 | 461.5 | 1.5 | **7.0** |
| | Deep Compression | 1.64 | 36.57 | 1.2 | 7.2 |
| Small VGG-16 | DeepCABAC | 0.95 | 63.2 | - | 9.0 |
| | MIRACLE | 0.16 | 375.0 | - | 10.0 |
| | EPR (DFT) | 0.10 | 600.0 | - | 10.0 |
| | **Our Method** | 0.09 | **641.5** | **4.7** | **7.0** |

Learning (MIRACLE) (Havasi et al. (2018)), Deep Compression (Han et al. (2016b)). Our experiments aim to examine the tradeoffs of our method on compression ratio, inference time and accuracy loss. In order to have a fair comparison, the input models for our method are pre-processed using OPQ method for Pruning and Quantization (Hu et al. (2021)), since other approaches also apply Pruning and Quantization methods (or other methods) before applying their methods.

Table 3 reports the results of this experiment. Our results show that our method can significantly outperform the state-of-the-art Model Coding methods on both compression ratio and speedup by a large magnitude, with the lowest loss of the accuracy. On ResNet-50, MobileNet-V2 and Small VGG-16, our method demonstrates the compression ratio up to 48.56×/30.11×/641.5× with conspicuous 9.2×/5.6×/4.7× speedup, with the minimum accuracy loss.

We make the key observation that: incorporating Pruning and Quantization with our method amplifies the benefits significantly to achieve the best outcome throughput the whole paper, compared with solely utilizing either Pruning or Quantization. On ResNet-50, our method outperforms the best results reported in Section 7.2 (i.e. 39.48× compression ratio, 4.7× speedup, 25.3% top-1 error), with 48.56× compression ratio, 9.2× speedup and only 24.5% top-1 error. Second, our method can enable lossless compression and acceleration at the same time. As shown in Table 3, there is no accuracy degradation for all models by using our method.

## 8    CONCLUSIONS

We present "Succinct Compression", a method to provide lossless compression of Deep Neural Network (DNN) models for fast and memory-efficient inference. Our method exploits *Succinct Data Structures* with three novel insights, which can be generally applicable to a variety of models (i.e. CNNs and Transformers). Our method is also synergistic with both structure-altered/-unaltered compress schemes (such as Pruning and Quantization). Our results justify the above benefits. On uncompressed models, our method provides at most 1.07× speedup with 1.17× compression ratio. The benefits from our method can be further amplified. By incorporating Pruning (and/or Quantization), we can achieve the maximum speedup/compression ratio at 9.3×/641.5×. We extensively examine and show that our method can significantly outperform six other Model Coding methods.

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

# A  ADDITIONAL DETAILS OF THE EXPERIMENTS

We disclose additional experimental details to make our work reproducible by others. More specifically, we focus on the detailed configurations of RAS types for different layers in the Deep Neural Network models, which is the most influential factor on the inference performance.

To this end, we provide additional details on how we decide the usage of different RAS and the quantitative supports, based on the requirement described in Section 3.3. Since our selection criteria is based on *FPr* (FLOPs-Parameter-ratio), we also provide relevant details for each layer.

Table A demonstrates the detailed results of this method on ResNet-50. As explained in our paper, the CONV layers demand Block-wise RAS for coarse-grained accesses, since accesses to these layers are relatively frequent; and the FC layers can be used via Element-wise RAS, since accesses to these layers are less frequent.

Table 4: **ResNet-50 MACs and RAS Type**

| Layers | #Parameters | MACs | MACs/#Parameters | RAS Type |
|---|---|---|---|---|
| conv1 | 9.41k | 118.01M | 12541 | Block-wise |
| layer1.conv | 215.81k | 680.39M | 3153 | Block-wise |
| layer2.conv | 1.22M | 1.04G | 852 | Block-wise |
| layer3.conv | 7.1M | 1.47G | 207 | Block-wise |
| layer4.conv | 14.96M | 811.02M | 54213 | Block-wise |
| fc | 2.05M | 2.05M | 1 | Element-wise |
| **Total** | **25.6M** | **4121.47M** | **161** | |

The same trend of the RAS distribution can also be found on VGG-16, where FC layers accounts for around 90% of all model parameters. Such an overwhelming proportion directly results in the fact that: the inference performance of our method on VGG-16 is significantly weakened, compared to the results on other models (where FC layers occupy less space).

Table 5: **VGG-16 MACs and RAS Type**

| Layers | #Parameters | MACs | MACs/#Parameters | RAS Type |
|---|---|---|---|---|
| conv1 | 1.79k | 89.92M | 50179 | Block-wise |
| conv2 | 36.93k | 1.85G | 50095 | Block-wise |
| conv3 | 73.86k | 926.45M | 12543 | Block-wise |
| conv4 | 147.58k | 1.85G | 12536 | Block-wise |
| conv5 | 295.17k | 925.65M | 3136 | Block-wise |
| conv6 | 590.08k | 1.85G | 3135 | Block-wise |
| conv7 | 590.08k | 1.85G | 3135 | Block-wise |
| conv8 | 1.18M | 925.25M | 784 | Block-wise |
| conv9 | 2.36M | 1.85G | 783898 | Block-wise |
| conv10 | 2.36M | 1.85G | 783898 | Block-wise |
| conv11 | 2.36M | 462.52M | 196 | Block-wise |
| conv12 | 2.36M | 462.52M | 196 | Block-wise |
| conv13 | 2.36M | 462.52M | 196 | Block-wise |
| fc1 | 102.76M | 102.76M | 1 | Element-wise |
| fc2 | 16.78M | 16.78M | 1 | Element-wise |
| fc3 | 4.1M | 4.1M | 1 | Element-wise |
| **Total** | **138.355M** | **15478M** | **112** | |

# B  INFERENCE ENGINE

Since the inference of our method requires extra supports to access more hardware-detailed functions, we extend PyTorch Paszke et al. (2019a) for our design. As shown in Figure 3, there are two parts in our extended version. First, we build extensions to support the harmonic transition from the tensor interface to our formalization. Second, we build a module to support runtime query, access and computation over Wavelet Tree Grossi et al. (2003).

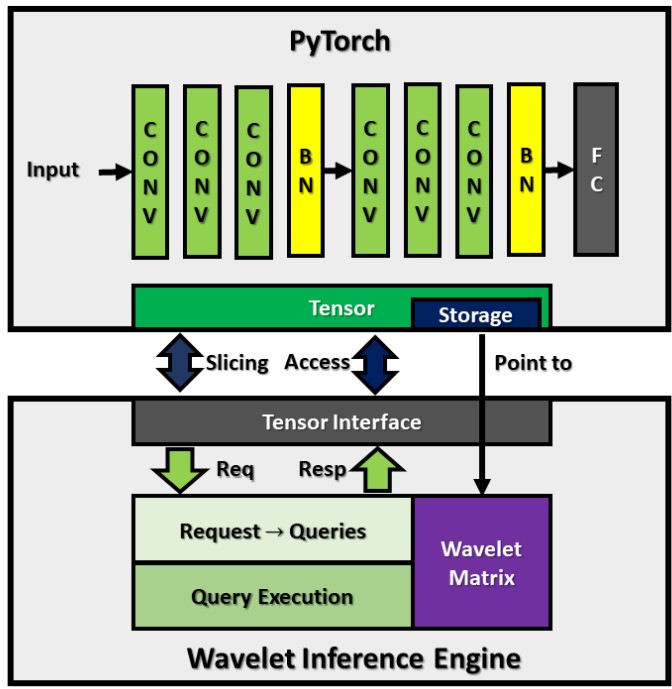

Figure 3: Wavelet Inference Engine

## B.1  TENSOR INTERFACE

We extend an override of the PyTorch tensor interface, for a new set of mechanisms including the storage, slice and data accesses. This guarantees the ease of usage from our method, since no extra efforts are needed. The rationles behind our extensions are as follows. First, the data storage format of PyTorch's tensor becomes inefficient when managing Succinct Data Structures: PyTorch separates the calculation logic and data storage for the tensor, and the storage is managed by a special module under the assumption that *all the data can be stored as a whole block of continuous memory*. So that the interpretation of these data (e.g., slice and view) is easy to be offloaded to tensor's implementation. Hereby, we describe how we can connect Succinct Data Structures with the tensor interface.

Slicing is an indispensable function of tensor manipulation, and we extend this function by replacing the original offset-based slicing with Wavelet Tree's *select* primitive. On our RAS formulation, select operation is natively equivalent to slicing. By leveraging the delimiters in RAS, *select* can locate any sub-parts under small time consumption. For instance, there is a slicing request $T[4][2][1]$ on a 4-dimension tensor $T$ whose size is $(128, 3, 224, 224)$. To response this request, we can execute 5 select operations in order to get the required range, which are $select_{d1}(5)$, $select_{d2}(3)$, $select_{d3}(2)$ and $select_{d3}(3)$ (where $d1$, $d2$ and $d3$ are the delimiters for dimension-1/2/3, respectively). Considering the equivalence between *select* and slicing, we substitute classic slicing with *select* operations in Succinct Data Structures.

Address-based access, which is another rudimentary operation supported by the tensor interface, is more difficult to be used for Wavelet Tree. Therefore, we override the original access function

with *access* operations in Succinct Data Structures. Unlike tensors (whose data are arranged continuously), data (in Wavelet Trees) are stored in a tree-like structure, with positions specifically identified by delimiters (e.g., Ian Munro et al. (2016)). To query data in Wavelet Tree, the direct approach is to recursively conduct $rank_0$ queries on each level of the bitmap Navarro (2014). However, as the size of bitmap grows (e.g., to 1MB), the query process becomes inefficient due to the pointer-chasing issues. We provide additional supports for this, by utilizing a combination of Wavelet Matrix Claude et al. (2015) and Accelerated Decoding Baruch et al. (2020) to accelerate access query on GPU (Described in the next section).

## B.2 WAVELET TREE QUERYING MODULE

As mentioned in Section B.1, in Succinct Data Structures, the query mechanism and data arrangement format are highly associated. Therefore, it's necessary to co-design them for the better performance. For data storage, rather than storing a tree-like structure which is the standard formulation of Wavelet Tree, we rearrange the Wavelet Tree to be formed in a SIMD-friendly format, 2D array. Compared to commonly-used linear data structure like 2D array, pointer-traced tree structures incur additional challenges in data transfer and runtime access Lu et al. (2017); NVIDIA (b). Therefore, the conventional approach for Wavelet Tree can't be directly accelerated on a GPU platform. To address this issue, we leverage Wavelet Matrix Claude et al. (2015), an equivalent of Wavelet tree but expressed as 2D array, for our implementation on GPU. As shown in Figure 4, our implementation of a Wavelet Tree now becomes a matrix with bitmaps, which, under the hood, can be stored in a 1D array of bytes. This linear data structure allows our method to exploit GPUs more effectively.

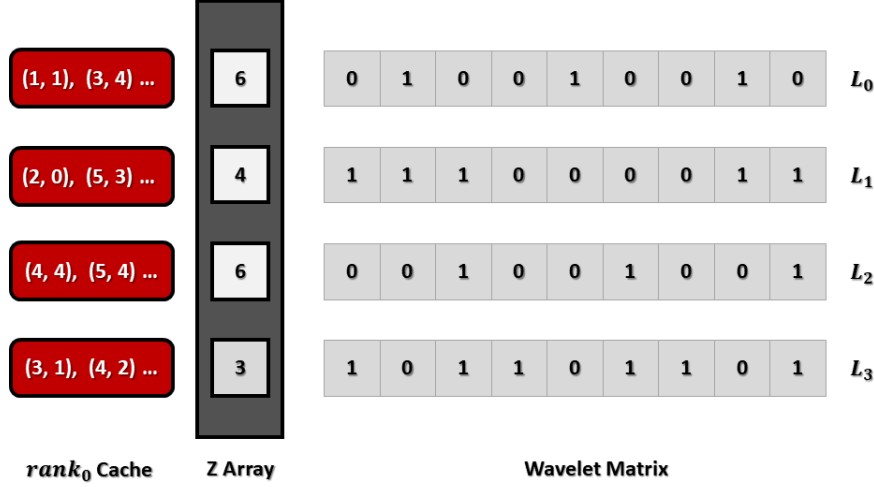

Figure 4: Wavelet Tree Query Module

To improve the performance of *rank* and *select*, we follow the methodology (described in Baruch et al. (2020)) to improve the performance. Our early characterization reveals that: $rank_0$ function [2] accounts for more than 90% computation time and is usually overlapped with other $rank_0$ calculations, if the system accesses on adjacent elements. Baruch et al. (2020) proposes a solution for this issue (though not addressed in DNN inference), by keeping the previous $rank_0$ result in a specific register. Our extensions allocate a block of cache for each layer, to facilitate with the reuse of previous results. We give out an pictorial example of this in Figure 4: the previous calculation results are stored as tuples in per layer cache; and, when the system invoke a new $rank_0$ function, the execution can directly start from the nearest recording. For a formal description of our algorithm, we provide the pseudo-codes in Algorithm 1.

---

[2] $rank_0$ is performed on each layer's bitmaps to count the number of zeros in a certain range.

---

**Algorithm 1** Access(x)

---

**Require:** $0 \leq x < the\ length\ of\ bitmap$
**Ensure:** $symbol,\ where\ symbol\ is\ the\ x_{th}\ element\ of\ original\ string$
  $l \leftarrow 0$
  $r \leftarrow len(\sigma)$                                             $\triangleright\ \sigma$ is the alphabet
  $level \leftarrow 0$
  $B \leftarrow WM[level]$                       $\triangleright\ WM$ is the Wavelet Matrix comprised of bitmaps
  **while** $r - l \neq 1$ **do**
      $(y, preResult) \leftarrow GetNearestRecording(x)$
      $q \leftarrow rank_0(B, y, x) + preResult$           $\triangleright\ rank_0$ counts the zeros between y and x
      $StoreRecording(x, q)$
      **if** $B[x] = 0$ **then**
          $x \leftarrow q$
          $r \leftarrow (r - l)/2$
      **else**
          $x \leftarrow Z[level] + (x - q)$        $\triangleright\ Z[level]$ is the number of zeros in current level
          $l \leftarrow (r - l)/2$
      **end if**
      $level \leftarrow level + 1$
      $B \leftarrow WM[level]$
  **end while**
  $symbol \leftarrow \sigma[l]$

---

