# OpenReview forum: "Succinct Compression: Lossless Compression for Fast and Memory-Efficient Deep Neural Network Inference"
_ICLR.cc/2023/Conference — Submitted to ICLR 2023_

### Official Review · Reviewer_PvEU · 2022-10-22

**Confidence:** 4
**Correctness:** 3
**Technical Novelty And Significance:** 2
**Empirical Novelty And Significance:** 3
**Recommendation:** 3

**Clarity, Quality, Novelty And Reproducibility:**

The technique part is difficult to track. It is casual in terms of lack of enough formulation. It is also not-intuitive that those design decisions are presented with enough justification.

The novelty of this paper is not clear.

There is a lack of code release for reproducibility or a plan to do so.

**Strength And Weaknesses:**

Strengthes:

+ Lossless compression method can be widely adopted in various inference workflows without considering the drawbacks of performance.

+ The combination with other lossy compression method is flexible.

Weaknesses:

- The presentation of the technique part is difficult to track. It is casual in terms of lack of enough formulation. It is also not-intuitive that those design decisions are presented with enough justification.

- The experimental section is limited to computer vision tasks, where the most memory intensive workflow is NLP models such as Bert and GPT in language modeling.

**Summary Of The Paper:**

This paper introduces a new data structure to support lossless compression of neural networks and show the effectiveness and efficiency of inference workflows in computer vision tasks.


**Summary Of The Review:**

The novelty of this paper is limited; the presentation of the technical section should be further polished; Although it shows a performance boost over computer vision tasks, they are inappropriate benchmarks to show the efficiency of model compression for inference based on the platform where the experiments have been conducted.

---

> ### Author Response · Authors · 2022-11-12
> **Thanks for your comments**
>
> ### Q1: The presentation of the technique part is difficult to track. It is casual in terms of lack of enough formulation. It is also not-intuitive that those design decisions are presented with enough justification.
> A1: We would be more than happy to revise, if the reviewer can be specific to any concerns. We assume other reviewers do not face the issue raised by this reviewer.
>
> ### Q2: The experimental section is limited to computer vision tasks, where the most memory intensive workflow is NLP models such as Bert and GPT in language modeling; Although it shows a performance boost over computer vision tasks, they are inappropriate benchmarks to show the efficiency of model compression for inference based on the platform where the experiments have been conducted.
> A2: We respectully disagree, since many representative model compression works (e.g. [1, 2, 3]) are all evaluated on computer vision tasks. We follow their methodology to design our experiments. Besides, our experiments also cover a transformer-based model DeiT-B. Since our method is not task-specific, it's expected to bring similar benefits for NLP models.
>
> ### Q3: There is a lack of code release for reproducibility or a plan to do so.
> A3: We add an appendix to increase the reproducibility of our work, and we also consider the release of our source codes (given the paper are fairly reviewed without any bias).
>
> ### Reference:
>
> [1] Song Han, Huizi Mao, and William J. Dally. Deep compression: Compressing deep neural network with pruning, trained quantization and huffman coding. In Yoshua Bengio and Yann LeCun (eds.), 4th International Conference on Learning Representations, ICLR 2016, San Juan, Puerto Rico, May 2-4, 2016, Conference Track Proceedings, 2016a.
>
> [2] Liu, Z., Li, J., Shen, Z., Huang, G., Yan, S., & Zhang, C. (2017). Learning efficient convolutional networks through network slimming. In Proceedings of the IEEE international conference on computer vision (pp. 2736-2744).
>
> [3] Zhuang Liu, Mingjie Sun, Tinghui Zhou, Gao Huang, and Trevor Darrell. Rethinking the value of network pruning, In 7th International Conference on Learning Representations, ICLR 2019.

---

### Official Review · Reviewer_BLqS · 2022-10-24

**Confidence:** 3
**Correctness:** 4
**Technical Novelty And Significance:** 3
**Empirical Novelty And Significance:** 3
**Recommendation:** 8

**Clarity, Quality, Novelty And Reproducibility:**

Clarity: fair. Needs more detailed steps of the algorithm
Quality: good
Novelty: good

**Strength And Weaknesses:**

## Strength
1. The proposed method can enable faster inference without influencing the model's performance.
2. It can be applied together with existing methods on model pruning and quantization.
3. The use of succinct data structure is well motivated.
4. Experiments are extensive and the speedup improvements are consistent.

## Weaknesses
1. The description of how to apply the succinct data structure is not very clear. Having an algorithm or pseudo code would be better.
2. It seems that certain software library is utilized for better performance of the proposed method on the hardware. This needs more detailed explanation. Are the reported baseline methods results also based on comparable libraries for performance/speed optimization?


**Summary Of The Paper:**

This paper proposes a novel method for lossless compression of DNNs which enables faster and memory-efficient inference. Specifically, it applies the succinct data structure to do the compression of DNNs. The main benefit of succinct data structure is that it supports queries without decompressing the compressed representations, thus being faster than previous model coding approaches. Experiments show better speedup ratio on both pruned models and quantized models with several different DNN architectures.

**Summary Of The Review:**

Overall the proposed method is novel and the experiments do support the speedup claim. Some details need more explanations.

---

> ### Author Response · Authors · 2022-11-12
> **Thanks for your comments**
>
> ### Q1: The description of how to apply the succinct data structure is not very clear. Having an algorithm or pseudo code would be better.
>
> A1: We add an appendix for more detailed information on our usage of the inference pipeline (by detailing the engineering choices, algorithms and pesudo codes, as suggested by the reviewer).
>
> ### Q2: It seems that certain software library is utilized for better performance of the proposed method on the hardware. This needs more detailed explanation. Are the reported baseline methods results also based on comparable libraries for performance/speed optimization?
>
> A2: We ensure the consistency of our method when comparing our method with the baseline.

---

### Official Review · Reviewer_MXqh · 2022-10-24

**Confidence:** 4
**Correctness:** 2
**Technical Novelty And Significance:** 2
**Empirical Novelty And Significance:** 3
**Recommendation:** 3

**Clarity, Quality, Novelty And Reproducibility:**

- This paper does not suggest new model compression algorithms that might be interesting to ML community.
- If the major contributions exist in data structures and a new formulate, it would be better to submit the manuscript to relevant conferences that handle those issues. The audience wanting to understand this manuscript would require background not very well known to ML community.
- Detailed profiling and analysis on how those CR and speed-up can be obtained should be provided. Especially for speed-up, results should be highly dependent on the hardware selection.
- Novelty seems to be quite limited.

**Strength And Weaknesses:**

*Strength
- It is interesting to apply new data structures to enable practical approaches of widely adopted model compression techniques, such as pruning and quantization.
- Compression ratio and inference speed can be improved by more than 1.2X and 4-9X, respectively, for ResNet-50, ResNet-100, and VGG DeiT-B.

*Weakness
- Detailed (mathematical) analysis on succinct data structures is missing while exploring (and comparing) other data structures needs to be included.
- Ablation study for Table 3 needs to be included. Since OPQ has been already applied as a pre-processing, it is difficult to estimate whether the overall improvement on CR or speedup is due to OPQ or RAS.
- What are the inherent advantages of RAS? How can RAS enhance memory saving?
- What is "specialized execution pipeline"? If it is based on NVIDIA CUDA Fortran, at least some introduction of such engines need to be introduced.


**Summary Of The Paper:**

The authors propose succinct data structures and a new formulate, Runtime-Accessible Sequence, to efficiently utilize new data structures. RAS is provided with two versions (element-wise and block-wise) with synergistic effects with pruning and quantization. The authors provide experimental results to support the claims with improved compression ratio and speedup.

**Summary Of The Review:**

- More background and related work need to be included.
- This reviewer is not sure whether ICLR is the right venue for this manuscript.
- Both mathematical and practical analysis need to be presented.
- What about other tasks, such as NLP? Is the proposed method specific to vision tasks?
- Overall, it is difficult for this reviewer to see how the proposed method can lead to "faster and more memory-efficient" inference.

---

> ### Author Response · Authors · 2022-11-12
> **Thanks for your comments**
>
> ### Q1: Detailed (mathematical) analysis on succinct data structures is missing while exploring (and comparing) other data structures needs to be included.
> A1: The mathematical advantage of Succinct Data Structures has been commonly-accepted in various prior works (e.g. [1] and [2]). Hence, we only give a short overview on the advantages.
>
> ### Q2: Ablation study of Table 3 is not conducted. Is the CR or speedup come from OPQ or RAS?
> A2: We clarify that RAS is only the 1D representation of model’s CONV weights, and not able to compress the weights. It is the succinct data structure (i.e. Wavelet Tree) applied to RAS compresses the weights. Besides, the ablation study results have already been shown in Table 3. The CR and speedup of OPQ are included in it. For instance, the OPQ on ResNet-50 brings about 1.3x speedup and 38x CR.
>
> ### Q3: What are the inherent advantages of RAS? How can RAS enhance memory saving?
> A3: RAS is the equivalent 1D representation of the 4D tensor of CONV weights, which will consume less memory space and enable the “select”, “rank” and “access” (from Succinct Data Structures). We re-explain the pipeline again: CONV weights (4D tensors) -> RAS (1D representations) -> succinct data structure (Wavelet Tree, which compresses RAS) -> conduct neural network inference on succinct data structure. Through viewing this pipeline, we can see that the compression is the result of applying succinct data structure on RAS rather than converting the 4D tensor to 1D RAS.
>
> ### Q4: What is "specialized execution pipeline"? If it is based on NVIDIA CUDA Fortran, at least some introduction of such engines need to be introduced.
> A4: It's our extensively-engineered version of PyTorch, to support our method. Please refer to the newly-added appendix in our revision.
>
> ### Q5: This paper does not suggest new model compression algorithms that might be interesting to ML community.
> A5: To the best of our knowledge, this is the first work to show that Succinct Data Structures can be used for ML compression and acceleration. Moreover, the synergy between our method and other model compression algorithms is expected to be interesting to ML community, which may stimulate further studies.
>
> ### Q6: If the major contributions exist in data structures and a new formulate, it would be better to submit the manuscript to relevant conferences that handle those issues. The audience wanting to understand this manuscript would require background not very well known to ML community.
> A6: We cover the neceesity of our work in Q5. Hereby, we clarify that: we don’t propose the new data structure Wavelet Tree, and we use it to succinctly compress the RAS. Besides, we believe our contributions are highly related to the important issues ML community concerns: how to further compress/accelerate the compressed models without any accuracy degradation. We assume the thriving state of ML community and ICLR comes from their great acceptance and respect for innovation from different viewpoints, and we believe our work indeed introduces a new perspective of model compression and acceleration for the first time.
>
> ### Q7: More background and related work need to be included.
> A7: We would add an additional section in appendix to cover this, if this paper is accepted.
>
> ### Q8: Detailed profiling and analysis on how those CR and speed-up can be obtained should be provided. Especially for speed-up, results should be highly dependent on the hardware selection.
> A8: Please refer to our answers for common questions.
>
> ### Q9: What about other tasks, such as NLP? Is the proposed method specific to vision tasks?
> A9: Our method is not task-specific, and it can be used for different models. Note that our experiments also contain the results by applying our method on a transformer-based model, DeiT-B.
>
> ### Reference:
> [1] Roberto Grossi, Ankur Gupta, and Jeffrey Scott Vitter. High-order entropy-compressed text indexes. In Proceedings of the Fourteenth Annual ACM-SIAM Symposium on Discrete Algorithms, SODA ’03, pp. 841–850, USA, 2003. Society for Industrial and Applied Mathematics. ISBN 0898715385.
>
> [2] Navarro, G. (2012). Wavelet Trees for All. In: Kärkkäinen, J., Stoye, J. (eds) Combinatorial Pattern Matching. CPM 2012. Lecture Notes in Computer Science, vol 7354. Springer, Berlin, Heidelberg. https://doi.org/10.1007/978-3-642-31265-6_2

---

### Official Review · Reviewer_DFkG · 2022-10-24

**Confidence:** 3
**Correctness:** 3
**Technical Novelty And Significance:** 4
**Empirical Novelty And Significance:** 4
**Recommendation:** 8

**Clarity, Quality, Novelty And Reproducibility:**

This paper has a good novelty to be the first to combine succinct data structure with nn inference, and also has comprehensive experimental studies to prove the effectiveness. It would be better to provide more descriptions on  theoretical justification and experimental details.

**Strength And Weaknesses:**

Strengths:
1. the paper combine a new data structure with the neural network inference and get good results.
2. the paper propose a flexible schema on where the data structures is used for neural network to further optimize the memory and time.
3. The benefits are verified on different types of popular neural networks including resnet and transformer.
Weaknesses:
1. Lack some descriptions on how this data structure is applied to neural networks? Are both the neural network weights and activations represented by the succinct data structure?
2. Can you provide more details on how to use pytorch with the succinct data structure? do you rewrite the pytorch kernels to directly use the succinct data structure, or there still need to convert to tensor representation before feeding into pytorch kernel?
3. The paper gives time/memory complexity of the succinct data structure, it will be better to compare this to the corresponding nn complexities for each kind of neural network(layer).
4. the speedup improvement on the uncompressed model is not significant compared to the compressed model?  Is there any formal explanation on this? such as vocabulary size change and so on?
5. It’s not very clear where the speedup comes from, like more efficient kernel, cache, data communication or anything else? Can you provide a theoretical analysis on why this new data structure runs faster compared to common nn libraries like pytorch if this is the first one to use it for nn? This might be related to issue 2.

**Summary Of The Paper:**

In  this paper,  the succinct data structures, which supports fast queries without decompressing the compressed representations, is utilized to do deep neural network inference. Also they propose a scheme to enable mixed-formulation inference for different layers.  The experimental results show that the proposed method not only can achieve better speedup and memory efficiency on both uncompressed and compressed models while preserving the accuracy, but also outperform the sota model coding approaches.


**Summary Of The Review:**

This paper proposed a good idea to apply the succinct data structure to the neural network inference. By combining with using the mixed schemas for different layers, it shows significant benefits on both time and memory requirements on different types of neural networks.  It would be better to provide more comparisons for the theoretical time/memory comparison instead of the experimental findings to justify when it will be suitable.

---

> ### Author Response · Authors · 2022-11-12
> **Thanks for your comments**
>
> ### Q1: Lack some descriptions on how this data structure is applied to neural networks? Are both the neural network weights and activations represented by the succinct data structure?
>
> A1: We use the succinct data structure to compress the weights. The weights in CONV layers are 4D tensors which can be converted to 1D RAS using the formulation method described in Section 3. We then use the succinct data structure to store 1D RAS.
>
> ### Q2: How to use pytorch with the succinct data structure?
>
> A2: We revise to add an appendix, to disclose the our engineering efforts.
>
> ### Q3: The paper gives time/memory complexity of the succinct data structure, it will be better to compare this to the corresponding nn complexities for each kind of neural network(layer).
>
> A3: It's desirable but we cannot derive the therotical results on DNN layers, since they can be highly dependent to the detailed numbers (which affect the compression ratio).
>
> ### Q4: The speedup improvement on the uncompressed model is not significant compared to the compressed model? Is there any formal explanation on this? such as vocabulary size change and so on?
>
> A4: This is due to the large alphabet. As mentioned in Section 4, the time complexity of query operations is the logarithm of vocabulary size. The uncompressed models contain a large amount of distinct symbols(numbers), and the overheads come from the construction phase also. Both contribute to the degradation of the inference speedup. We later show that combining other compression methods can unveil the benefits of our method, by reducing overheads from the the above two parts.
>
> ### Q5: It’s not very clear where the speedup comes from, like more efficient kernel, cache, data communication or anything else? Can you provide a theoretical analysis on why this new data structure runs faster compared to common nn libraries like pytorch if this is the first one to use it for nn?
>
> A5: (1) For the first question, please refer to our answer for common questions.
> (2) As answered in Q3, we assume we can not deliver a theoretical analysis on this point.

---

### Author Response · Authors · 2022-11-12
**Revision Summary**

We thank all reviewers for the suggestions, and provide a revision hereby.

The revision adds a new appendix, which discloses the details on how we build the specialized execution pipeline for the inference over Succinct Data Structures.

---

### Author Response · Authors · 2022-11-12
**Response to General Questions**

We thank all reviewers for their constructive and insightful feedback. We are particularly excited to see Reviewer DFkG and BLqS highly appreciate the values of our work. Hereby, we clarify some general questions raised by the reviewers.

### Q1: Where does the speedup come from? [Reviewer DFkG and MXqh]
A1: Our speedup comes from three aspects. First, our method further compresses the model, which reduces the host-accelerator transfer overheads; second, our method reduces the opearting bit-width, which allows more data can be pushed within the on-chip caches; and third, our method performs operand unpacking (using low-level hardware intrinsic functions via Fortan), which allows us to unpack packed operands (e.g. eight 4-bit operands in a 32-bit packed one) efficiently (by going out to the cache/memory as less as possible).

### Q2: Could you provide a more detailed description of your inference pipeline? [Reviewer DFkG, MXqh and BLqS]
A2: We revise to add the appendix to include the design details of our pipeline.

### Q3: Can Succinct Compression be applied to NLP? [Reviewer MXqh and PvEU]
A3: Our method is not task-specific, and our experiments also contain the performance evaluation of applying our method on a transformer-based model, DeiT-B. Besides, we disagree with Reviewer PvEU’s statement that “computer vision tasks are inappropriate benchmarks”. Many representative model compression works like [5, 6, 7] are all evaluated on computer vision tasks. We follow their methodology to design our experiments.

### Q4: Novelty is limited. [Reviewer MXqh and PvEU]
A4: Our work is the first work to introduce succinct data structure to model compression, which achieves great and consistent compression and speedup on (un)compressed models.


### Reference:
[1] Francisco Claude, Gonzalo Navarro, and Alberto Ord ́o ̃nez. The wavelet matrix: An efficient wavelet tree for large alphabets. Information Systems, 47:15–32, 2015. ISSN 0306-4379. doi:https://doi.org/10.1016/j.is.2014.06.002.

[2] Gilad Baruch, Shmuel T. Klein, and Dana Shapira. Accelerated partial decoding in wavelet trees. Discrete Applied Mathematics, 274:2–10, 2020. ISSN 0166-218X. doi:https://doi.org/10.1016/j.dam.2018.07.016.

[3] Steven K. Esser, Jeffrey L. McKinstry, Deepika Bablani, Rathinakumar Appuswamy, and Dharmendra S. Modha. Learned step size quantization, In 8th International Conference on Learning Representations, ICLR 2020.

[4] Liu, Z., Wang, Y., Han, K., Zhang, W., Ma, S., & Gao, W. (2021). Post-training quantization for vision transformer. Advances in Neural Information Processing Systems, 34, 28092-28103.

[5] Song Han, Huizi Mao, and William J. Dally. Deep compression: Compressing deep neural network with pruning, trained quantization and huffman coding. In Yoshua Bengio and Yann LeCun (eds.), 4th International Conference on Learning Representations, ICLR 2016, San Juan, Puerto Rico, May 2-4, 2016, Conference Track Proceedings, 2016a.

[6] Liu, Z., Li, J., Shen, Z., Huang, G., Yan, S., & Zhang, C. (2017). Learning efficient convolutional networks through network slimming. In Proceedings of the IEEE international conference on computer vision (pp. 2736-2744).

[7] Zhuang Liu, Mingjie Sun, Tinghui Zhou, Gao Huang, and Trevor Darrell. Rethinking the value of network pruning, In 7th International Conference on Learning Representations, ICLR 2019.

---

### Decision · Program_Chairs · 2023-01-20

**Decision:**

Reject

**Justification For Why Not Higher Score:**

The paper lacks sufficient clarity to be understood by a general audience and there is a lack of confidence that this could be reproduced.

**Justification For Why Not Lower Score:**

N/A

**Metareview: Summary, Strengths And Weaknesses:**

The paper proposes succinct data structures for lossless compression of network models for fast and memory-efficient inference.  Whilst the paper seems interesting, most of the reviewers struggled to understand the details of the work, even after the discussion phase. In general the consensus is that the work suffers from a lack of clarity and reproducibility. The work in that sense isn't yet ready for presentation at ICLR.

**Summary Of Ac-Reviewer Meeting:**

We had extensive discussions online and the two reviewers who scored this initially at 8 said they would reduce their scores to 6 and 5. For some reason they didn't actually input these new scores (perhaps they assumed the AC would do it). With these updated scores, the paper is significantly below threshold. This is the correct decision I feel in this case -- the paper is almost unintelligible in places.